

**Impact of fossil and non-fossil sources on the molecular compositions of water soluble humic-**
**like substance in PM$_{2.5}$ at a suburb site of Yangtze River Delta, China**
Mengying Bao[1,2,3], Yan-Lin Zhang[1,2,*], Fang Cao[1,2], Yihang Hong[1,2], Yu-Chi Lin[1,2], Mingyuan Yu[1,2],
Hongxing Jiang[4,5], Zhineng Cheng[4,5], Xiaoying Yang[1,2]
*1 School of Applied Meteorology, Nanjing University of Information Science & Technology,*
*Nanjing 210044, China.*
*2 Atmospheric Environment Center, Joint Laboratory for International Cooperation on Climate*
*and Environmental Change, Ministry of Education (ILCEC), Nanjing University of Information*
*Science & Technology, Nanjing 210044, China.*
*3 Huzhou Meteorological Administration, Huzhou 313300, China*
*4 State Key Laboratory of Organic Geochemistry and Guangdong province Key Laboratory of*
*Environmental Protection and Resources Utilization, Guangzhou Institute of Geochemistry,*
*Chinese Academy of Sciences, Guangzhou 510640, China.*
*5 CAS Center for Excellence in Deep Earth Science, Guangzhou 510640, China*
*Correspondence: Yan-Lin Zhang (dryanlinzhang@outlook.com)*
**Abstract**
Atmospheric humic-like substances (HULIS) affect global radiation balance due to its strong
light absorption at the ultraviolet wavelength. The potential sources and molecular compositions
of water soluble HULIS at a suburb site of Yangtze River Delta from 2017 to 2018 were discussed
based on the radiocarbon ($^{14}$C) analysis combining the Fourier Transform Ion Cyclotron
Resonance Mass Spectrometry (FT-ICR MS) technique in this study. The $^{14}$C results showed that
the averaged non-fossil source contributions to HULIS were 39 ± 8 % and 36 ± 6 % in summer
and winter, respectively, indicating that both the fossil and non-fossil sources played important
roles in the formation of HULIS. The Van Krevelen diagrams obtained from the FT-ICR MS
results showed that the proportions of tannins-like and carbohydrates-like groups were higher in
summer, suggesting significant contribution of HULIS from biogenic secondary organic aerosols
(SOA). The higher proportions of condensed aromatic structures in winter suggested the increasing
anthropogenic emissions. Molecular composition analysis on the CHO, CHON, CHOS, and



CHONS subgroups showed the relatively higher intensities of high O-containing macromolecular
oligomers in CHO compounds in summer, further indicating stronger biogenic SOA formation in
summer. High-intensity phenolic substances and flavonoids which were related to biomass burning
and polycyclic aromatic hydrocarbons (PAHs) derivatives indicating fossil fuel combustion
emissions were found in winter CHO compounds. Besides, two high-intensity CHO compounds
containing condensed aromatic ring structures ($C_9H_6O_7$ and $C_{10}H_5O_8$) identified in summer and
winter samples were similar to those from off-road engine samples, indicating that traffic emission
was one of the important fossil sources of HULIS at the study site. The CHON compounds were
mainly composed of organonitrates or nitro compounds with significant higher intensities in winter,
which was associated to enhanced formation of organonitrates due to high $NO_x$ in winter. However,
the high-intensity CHON molecular formulas in summer were referring to N-heterocyclic aromatic
compounds, which were produced from the atmospheric secondary processes involving reduced
N species (e.g., ammonium). The S-containing compounds were mainly composed of
organosulfates (OSs) derived from biogenic precursors, long-chain alkane and aromatic
hydrocarbon, further illustrating the mixed sources of HULIS and both important biogenic and
anthropogenic source contributions to HULIS at the study site.  These findings add to our
understanding of the interaction between the sources and the molecular compositions of
atmospheric HULIS.

**1. Introduction**

Atmospheric humic-like substances (HULIS) have been observed worldwide and can be
produced from primary combustion of biomass, fossil fuel, as well as various secondary processes
such as photochemical processes of volatile organic compounds (VOCs) and heterogeneous
reactions of organic aerosols in the atmosphere (Kuang et al., 2015; Li et al., 2019; Ma et al., 2018;
Sun et al., 2021). As important component of brown caron (BrC) aerosols, HULIS species have
been widely reported to have a great impact on global radiative budget, contributing to 20-40% of
the direct radiative forcing caused by light absorbing aerosols due to its light absorption at the
ultraviolet wavelength (Chung et al., 2012; Zhang et al., 2017; Zhang et al., 2020; Wang et al.,
2018b). HULIS are a highly complex mixture of polar organic compounds composed of aromatic
and hydrophobic aliphatic structures containing carboxyl, carbonyl, and hydroxyl function groups
(Zheng et al., 2013; Graber and Rudich, 2006). During the atmospheric secondary oxidation





processes, the substitutions of hydrophilic functional groups produced aerosol hygroscopicity (Huo et al., 2021; Jiang et al., 2020). Polycarboxylic acids in HULIS are surface-active and play an important role in the cloud condensation nuclei (CCN) activity (Tsui and McNeill, 2018). N-base compounds can promote the generation of atmospheric reactive oxygen species (ROS) which have a great impact on human health (Wang et al., 2017b; De Haan et al., 2018; Song et al., 2022). Identifying the molecular compositions of HULIS is a challenge due to complex mixtures contained in HULIS and can help to a better understanding of the processes involving organic compounds in atmosphere (Noziere et al., 2015; Laskin et al., 2018).

The Fourier-Transform Ion Cyclotron Resonance Mass Spectrometry (FT-ICR MS) coupled with electrospray ionization (ESI) ion source have been widely used in identifying the chemical structure of HULIS, providing high mass accuracy and can determine molecular formulas from mixed compounds (Chen et al., 2016; Wang et al., 2019b; Lin et al., 2012a; Jiang et al., 2020). Typical molecular formulas composed of C, H, and O atoms in HULIS were observed being abundant in carboxylic acids, lingin-derived products, and polycyclic aromatic hydrocarbons (PAHs) or their derivatives (Lin et al., 2012a; Sun et al., 2021; Jiang et al., 2020; Huo et al., 2021; Song et al., 2018). In addition, the HULIS formation of N and S containing precursors were also widely detected. The N-containing compounds such as nitroaromatics were important chromophores in HULIS in aged biomass burning organic aerosols (BBOA), as well as in ambient aerosols influenced by biomass burning (BB), while reduced N compounds such as N-heterocyclic aromatic compounds were found to be important chromophores in fresh BBOA (Wang et al., 2019b; Song et al., 2022; Jiang et al., 2020; Wang et al., 2017b). Recent laboratory simulation experiments showed that the photooxidation of various anthropogenic VOCs (e.g., naphthalene, benzene, toluene, and ethylbenzene) would be promoted under high $NO_x$ condition, producing strongly light absorbing nitroaromatics (Yang et al., 2022; Aiona et al., 2018; Siemens et al., 2022; Xie et al., 2017). Otherwise, nighttime oxidation of biogenic or anthropogenic VOCs, such as benzene/toluene, isoprene ($C_5H_8$) and monoterpenes ($C_{10}H_{16}$) by $NO_3$ radicals lead to substantial organonitrates formation, where the VOCs oxidation is strongly affected by $NO_x$ (He et al., 2021; Shen et al., 2021; Wang et al., 2020; Zheng et al., 2021).

The organosulfates (OSs) and nitrooxy organosulfates (nitrooxy-OSs) have also been found to widely exist in HULIS in different atmospheric environment (Lin et al., 2012b; Lin et al., 2012a; Sun et al., 2021). Field study and laboratory smog chamber experiments have confirmed that OSs





and nitrooxy-OSs in the atmosphere mainly come from the $O_3$, OH, or $NO_3$ oxidation of biogenic
VOCs such as isoprene, α/β-pinene as well as aromatic hydrocarbon in the presence of $H_2SO_4/SO_2$
(Surratt et al., 2008; Glasius et al., 2021; Yang et al., 2020; Lin et al., 2012b; Huang et al., 2020).
Coal combustions were found to be important sources of the aromatic OSs and nitrooxy-OSs in
HULIS (Song et al., 2018). Besides, the long-chain alkanes were found to be important precursor
of OSs in atmospheric aerosol samples from urban area which was related to vehicle emissions
(Wang et al., 2019a; Tao et al., 2014).

Nanjing is one of the main cities in the Yangtze River Delta (YRD), which is one of the most

developed areas in China. Previous study reported by our laboratory have found significant HULIS
formation at Nanjing, China influenced by both biogenic and anthropogenic emissions (Bao et al.,
2022). The molecular compositions of water soluble HULIS isolated from $PM_{2.5}$ samples collected
in summertime and wintertime from 2017 to 2018 at Nanjing, China, were investigated combining
the FT-ICR MS and radiocarbon ($^{14}$C) analysis. We aim to obtain the molecular characteristic
differences of water soluble HULIS in summertime and wintertime and to get a better
understanding of the influence of different sources on the molecular compositions of HULIS.
**2. Materials and methods**
2.1 Sample collection

The 24 h $PM_{2.5}$ samples were collected on the roof of Wende building, which was about 21

m height from the ground at Nanjing University of Information Science and Technology (32.2° N,
118.7° E) using a high-volume sampler (KC-1000, Qingdao, China) at a flow rate of 300 L min$^{-1}$.
The study site was located at the north suburb area of Nanjing, adjacent to G205 State Road and
surrounded by an industrial park and residential area. Generally, the study site was affected by
human activity, industrial emission, and traffic emission. The sample collection was conducted in
summer from 12 August 2017 to 26 August 2017 and in winter from 31 December 2017 to 31
January 2018. A heavy haze event occurred from 31 December 2017 to 3 January 2018, thus the
sample frequency was adjusted to 2 h in daytime and 8 h in nighttime. Field blank filters were
performed before and after sample collection for each season. More details about the sample
collection can be found in previous research reported by our laboratory (Bao et al., 2022). The air
pollutants data were provided by China National Environmental Monitoring Centre. Twelve
samples were selected for further chemical analysis and the details about the sample selection are
described in Section 3.1 in this study.





2.2 Chemical analysis

The solid phase extraction (SPE) was performed to isolate the water soluble HULIS in this study and the carbon fraction in HULIS (HULIS-C) were determinated using a total carbon analyzer (Shimadzu-TOC-VCPH, Shimadzu, Japan) with standard deviation of reproducibility test less than 3.5% and detection limit of 0.14 µg C m$^{-3}$. The mass concentrations of levoglucosan were measured using an ion chromatography (Dionex ICS-5000+, ThermoFisher Scientific, USA) and the details of the methods have been described previously (Liu et al., 2019). All data were blank corrected in this study. More details about the HULIS isolation and measurement have been described in (Bao et al., 2022).

2.3 Radiocarbon analysis

For the radiocarbon measurement of the HULIS samples, the organic solvents were firstly evaporated under a gentle flow of ultrapure $N_2$ for 30-40 minutes in tin cups. After that, the tin cups were wrapped into balls and more than 50 µg of carbon from the HULIS samples was combusted into $CO_2$ using an elemental analyzer (EA, model vario micro, elemental, Germany), then reduced into graphite targets for $^{14}C$ determination at the State Key Laboratory of Organic Geochemistry, Guangzhou Institute of Geochemistry, Guangzhou, China (Jiang et al., 2020). Detailed descriptions of the $^{14}C$ data processing can be found in previous study (Mo et al., 2018). Briefly, the $^{14}C$ values were expressed as the modern carbon ($f_m$) fraction after correcting for the $\delta^{13}C$ fractionation. The $f_m$ was converted into non-fossil carbon ($f_{nf}$) fraction with the correction factor of 1.06±0.07 based on the long-term time series of $^{14}CO_2$ sampled at the background station in this study (Levin et al., 2013; Levin and Kromer, 2004). No field blank correction was performed for the carbon isotope analysis since the carbon content in the field blanks was negligible.

2.4 High-resolution FT-ICR MS analysis

The ultrahigh resolution mass spectra of the HULIS samples were obtained through a SolariX XR FT-ICR MS (Bruker Daltonics, GmbH, Bremen, Germany) equipped with a 9.4 T superconducting magnet (Gamry Instruments, Warminster, USA) and a Paracell analyzer cell (Brucker Daltonik GmbH, Bremen, Germany) in the negative ESI mode. The detection mass range was set as m/z 150 to 800 and the ion accumulation time was set as 0.65 s. A total of 100 continuous 4M transient data points were superposed to enhance the signal to noise ratio and dynamic range. The mass spectrum was externally calibrated with a standard solution of arginine and internal





recalibration was performed using typical $O_6S_1$ chemical species in DataAnalysis ver. 4.4 software
(Bruker Daltonics) (Mo et al., 2018; Tang et al., 2020; Jiang et al., 2020). Field blank filters were
analyzed as same as the samples and all the sample data were blank corrected. More details about
the data processing can be found in Text S1 in the supporting information.
**3. Results and discussion**
3.1 General temporal characteristics during the sampling periods

Figure 1 displays the temporal variations of non-fossil contributions to HULIS-C, the mass

concentrations of HULIS-C, levoglucosan, $NO_3^-$, $SO_4^{2-}$, $NH_4^+$, $SO_2$, $NO_2$, and $PM_{2.5}$, as well as the
relative humidity and temperature during the study periods corresponding to the 12 samples. The
12 samples were named as S1-S6 and W1-W6 in chronological order in this study. The averaged
mass concentrations of $PM_{2.5}$ in summer and winter during the selected periods were $21.05 \pm 8.05$
$\mu g\ m^{-3}$ and $445.67 \pm 275.00\ \mu g\ m^{-3}$, respectively, indicating the serious pollution level in winter.
The daily $PM_{2.5}$ mass concentrations in summer were all below the daily averaged Chinese
National Ambient Air Quality Standard (NAAQS) of $35\ \mu g\ m^{-3}$ for the first grade, while the daily
$PM_{2.5}$ mass concentrations in winter all exceeded the daily averaged NAAQS of $35\ \mu g\ m^{-3}$ for the
first grade, of which the $PM_{2.5}$ mass concentrations of W1-W3 and W6 exceeded $200\ \mu g\ m^{-3}$.

As shown in Fig. 1, the mass concentrations of HULIS-C, levoglucosan, water soluble

secondary inorganic aerosols (SIA), and air pollutants showed similar trends in winter, suggesting
the influence of BB and anthropogenic emissions in winter (Wu et al., 2019b). Significant
increasing of the levoglucosan and HULIS-C mass concentrations were found from 31 December
2017 to 1 January 2018, corresponding to the W1-W3 samples, indicating the BB impact during
the periods. The maximum of the levoglucosan and HULIS-C mass concentrations were 552.79
$\mu g\ m^{-3}$ and $7.40\ \mu g\ m^{-3}$, respectively. Despite the higher levoglucosan mass concentrations in the
W1-W3 samples, the radiocarbon analysis results showed that the $f_{nf}$ of HULIS-C ranged from 30 %
to 50 % with an average contribution of $39 \pm 8$ % in summer and ranged from 32 % to 48 % with
an average contribution of $36 \pm 6$ % in winter, indicating that both fossil and non-fossil sources
played important roles in the formation of HULIS at the study site. There were other emission
sources of HULIS in winter other than BB. Figure S1 shows the 48 h back trajectories of each
sample during the selected periods. The study site was affected by the clean air masses from the
ocean in summer and the air masses mainly from the northern heating cities in winter, suggesting
the coal combustion contributions to HULIS in winter.





3.2 Mass spectra and molecular formula assignments

Figure S2 and S3 shows the negative ion ESI FT-ICR mass spectra of HULIS in summer and

winter, respectively. The molecular formulas listed are some of the top ten molecular formulas.
Thousands of peaks are present in the spectra in the range from m/z 150 to m/z 600 and the most
intense ion peaks are those in the range m/z 200-400 in summer and m/z 150-350 in winter. Our
results are similar to those found for the ultrahigh resolution mass spectra of water-soluble organic
compounds in particles produced from BB, coal combustion, vehicle exhaust emissions, as well as
in ambient aerosols and cloud water samples (Tang et al., 2020; Sun et al., 2021; Song et al., 2018;
Song et al., 2019; Bianco et al., 2018). In this study, the assigned molecular formulas were
classified into the following four main subgroups based on their elemental compositions: CHO
(compounds containing only C, H, and O), CHON (compounds containing C, H, O and N), CHOS
(compounds containing C, H, O, and S), and CHONS (compounds containing C, H, O, N, and S).
As shown in Fig. 2, the proportions of the four subgroups accounted for the overall formulas
followed as CHO (20 %-27 %), CHON (28 %-43 %), CHOS (19 %-26 %), and CHONS (16 %-
26 %) in summer, respectively and CHO (15 %-19 %), CHON (30 %-40 %), CHOS (21 %-32 %),
and CHONS (20 %-29 %) in winter, respectively. The average proportions of the CHO, CHON,
CHOS, and CHONS compounds in summer were $22 \pm 3$ %, $36 \pm 5$ %, $22 \pm 3$ %, and $20 \pm 4$ %,
respectively. The average proportions of the four subgroups in winter were $17 \pm 2$ %, $32 \pm 4$ %,
$24 \pm 3$ %, and $27 \pm 4$ %, respectively. The CHON groups were the major components of molecular
formulas. Notably, the contributions of S-containing compounds (CHOS and CHONS groups)
increased in winter which might be related to the polluted air masses transported from the northern
heating cities with increasing coal combustions emissions in winter (Song et al., 2018).

Table S1 and S2 displays the composition characteristics of atmospheric HULIS in the

summer and winter samples, including the relative intensity weighted average values of number,
molecular weight ($MW_w$), elemental ratios ($O/C_w$ and $H/C_w$), double-bond equivalent ($DBE_w$),
aromaticity index ($AI_w$), and $DBE/C_w$. A total of 14387 and 15731 peaks were detected in the
summer and winter samples, respectively. The O/C and H/C ratios are commonly calculated to
evaluate the oxidation degree and saturation degree of the compounds, respectively (Ning et al.,
2022). The $O/C_w$ values were in a range of 0.61-0.80 with an average value of $0.71 \pm 0.07$ for
summer samples and in a range of 0.59-0.67 with an average value of $0.62 \pm 0.03$ for winter
samples, respectively. The higher oxidation degree of summer samples than winter samples



indicated stronger secondary HULIS formation in summer. The $H/C_w$ values were in a range of
1.38-1.46 with an average value of $1.42 \pm 0.03$ for summer samples and in a range of 1.33-1.41
with an average value of $1.36 \pm 0.04$ for winter samples, respectively. The $O/C_w$ and $H/C_w$ of each
molecular subgroup followed a changing trend of CHO < CHON < CHOS < CHONS compounds.
Most of the S-containing compounds had a O/C value $\geq 0.7$, suggesting the large amounts of highly
oxidized OSs in S-containing compounds which contained various functional groups and were
mainly from the photochemical oxidation of biogenic or anthropogenic volatile organic
compounds (VOCs) (Mutzel et al., 2015). The DBE values were calculated to describe the degree
of unsaturation of compounds and restricted the assigned molecular formulas with unreasonably
high or low number of rings or double bonds (Kroll et al., 2011). The higher $DBE_w$ and $DBE/C_w$
values of CHO and CHON compounds were found in this study.

Considering that double bonds can be formed by heteroatoms especially O atoms, whereas

make no contributions to the aromaticity of the compounds, $AI_w$ was calculated to supplement the
DBE results (Song et al., 2018; Ning et al., 2019). $AI_w$ can eliminate the contribution of O, N, and
S atoms to the C=C double bond density of molecules. The $AI_w$ values of different compounds
groups in HULIS presented the changing trends: $AI_w$ (CHONS) > $AI_w$ (CHON) > $AI_w$ (CHO) >
$AI_w$ (CHOS) in summer and $AI_w$ (CHON) > $AI_w$ (CHO) > $AI_w$ (CHONS) > $AI_w$ (CHOS) in winter,
respectively. The formulas can be classified into three parts based on AI values proposed by
previous studies: aliphatic (AI =0), olefinic (0< AI $\leq$0.5) and aromatic (AI >0.5) (Koch and Dittmar,
2006). As shown in Fig. S4 and S5, the aliphatic were the main components of S-containing
compounds in this study and the olefinic and aromatic were the main components of CHO and
CHON compounds. Furthermore, the aromatic proportion of CHO and CHON compounds
significantly increased in winter, suggesting the increasing anthropogenic emissions in winter.
3.3 Comparative analysis using Van Krevelen diagrams

In this study, the Van Krevelen diagrams (Fig. 3) were constructed to display the molecular

composition and categorical distribution of the collected samples (Noziere et al., 2015; Patriarca
et al., 2018; Li et al., 2022). According to the elemental ratios (O/C and H/C ratios) and AI values,
seven major compound classes were classified, including lipids-like species, lignins-like species,
proteins-like species, tannins-like species, carbohydrates-like species, condensed aromatics
structure, and unsaturated hydrocarbons (Table S3). The Van Krevelen diagrams showed similar
distributions in the 12 samples. The CHO and CHON compounds located in the lower left area



and the S-containing compounds located in the upper light area with higher O/C and H/C ratios,
indicating a higher degree of oxidation and saturation. The condensed aromatic structure mainly
consisted in CHO and CHON compounds, further suggesting the influence of anthropogenic
emissions on the formation of CHO and CHON compounds.

Figure 4 presents the averaged relative contributions of the number of molecular formulas

from the seven categories in summer and winter samples, respectively. Lignins-like species
accounted for the highest proportion of CHO compounds with average contributions of 58 % and
61 % in summer and winter, respectively, followed by CHON compounds with average
contributions of 48 % and 57 % in summer and winter, respectively. Lignins are mainly composed
of carboxyl groups, alicyclic rings, aromatic rings, and other O-containing groups. Previous studies
have reported that lignin was a complex phenolic polymer which usually came from direct
biological emissions or combustions of biofuel (Ning et al., 2019; Boreddy et al., 2021; Sun et al.,
2021). Lignins pyrolysis products and other lignins derived molecules have been shown to be
oxidized into light absorbing BrC chromophore under certain conditions (Fleming et al., 2020).

Tannins-like species accounted for 21 %, 27 %, 23 %, and 30 % of CHO, CHON, CHOS, and

CHONS compounds, respectively in summer which were higher than those in winter with
contributions of 13 %, 16 %, 16 %, and 23 % to CHO, CHON, CHOS, and CHONS compounds,
respectively. Tannins-like species are a series of polyphenolic compounds containing hydroxyls
and carboxylic groups which have been widely reported in fogs, cloud water and aerosol samples,
attributing to highly oxidized organic compounds such as OSs or nitrooxy-OSs produced from the
nighttime chemistry between the biogenic VOCs with the $NO_3$ (Altieri et al., 2009; Bianco et al.,
2018; Ning et al., 2019; Altieri et al., 2008; Shen et al., 2021). Carbohydrates-like species which
contain monosaccharide, alditols, and anhydrosugars mainly consisted in CHONS compounds
which also had a relative higher proportion of 33 % in summer than that of 29 % in winter (Sun et
al., 2021). $C_{10}H_{16}NO_{7-9}S$, as monoterpene nitrooxy-OSs, showing high relative intensities, were
typical carbohydrates-like species detected in this study which represented biogenic secondary
organic aerosols (SOA) (Ning et al., 2019; Surratt et al., 2008; Wang et al., 2020). Both the higher
proportions of tannins-like and carbohydrates-like classes in summer indicated stronger biogenic
SOA formation in this study.

Proteins-like classes mainly consisted in CHOS compounds with average proportions of 29 %

and 38 % in summer and winter, respectively. Proteins contain peptide-like structures formed by





dehydration with different kinds of amino acids and consist of short chains of amino acid residues
(Bianco et al., 2018). These compounds are associated with photochemical oxidation processing
in aerosols, thus resulting in the significant formation of OSs from biogenic or anthropogenic
precursors in this study (Bigg and Leck, 2008).
Higher condensed aromatics were detected in winter with average proportions of 14 % in
CHO compounds and 8 % in CHON compounds, respectively which were 2-2.5 times of those in
summer. Condensed aromatics are important components of PAHs which usually emitted from
incomplete combustion of fossil fuels (Ma et al., 2020). The increase of the proportion of
condensed aromatics in winter indicated the stronger influence of anthropogenic sources on
HULIS formation. The unsaturated hydrocarbons and lipids-like species showed the lowest
molecular number percentage of less than 1 % in this study. Previous studies have shown that the
lipids-like species were the main components of water insoluble organic compounds in aerosols
and could be attributed to monocarboxylic acids (Ning et al., 2022; Wozniak et al., 2008).
In summary, both the summer and winter samples were mainly composed of compounds from
biogenic origins (lignins-like, tannins-like, proteins-like, and carbohydrates-like species). It is
noted that ESI ionization technology is more sensitive for the identification of polar compounds.
Therefore, the low polar or nonpolar compounds, such as PAHs or their derivatives from fossil
sources, were probably underestimated in this study (Jiang et al., 2014; Lin et al., 2018).
3.4 Molecular composition of HULIS
3.4.1 Molecular characteristics of CHO compounds
The $O/C_w$ and $H/C_w$ ratios for the CHO compounds were 0.45-0.56 and 1.15-1.30 for the
summer samples and 0.42-0.48 and 0.90-1.02 for the winter samples (Table S1 and S2). The
summer samples showed higher oxidation degree and saturation degree. We firstly plotted the Van
Krevelen diagrams of the four molecular subgroups showing relative intensities for all the 12
samples and similar distributions of the high-intensity compounds were found in the 6 summer
samples and the 6 winter samples, respectively. Then we combined all the data in summer and
winter, respectively. As shown in Fig. 5a and 5d, the CHO compounds in summer with high
relative abundance were located at the area within $0.2 \leq O/C \leq 1.0$ and $1.0 \leq H/C \leq 1.7$, mainly
including lignins-like species and tannins-like species which were closely related to biogenic
emissions. On the contrary, the condensed aromatics showed high relative abundance in winter,
suggesting the obvious different sources of HULIS in summer and winter. The DBE values





increased with the increasing of the C numbers (Fig. 5b and 5e). The high-intensity CHO
compounds in HULIS had DBE values between 3-7 with C numbers from 10 to 20 for summer
samples. In winter, the high-intensity CHO compounds had DBE values between 7-11 with C
numbers from 5 to 15. As mentioned above, the aromatic (AI >0.5) proportion of CHO compounds
significantly increased in winter, the higher DBE values in winter further indicated the consists of
more highly unsaturated aromatic compounds which reflected the anthropogenic emissions.

The CHO compounds were classified according to the number of oxygen atoms to evaluate

the oxygen content. As shown in Fig. 5c and 5f, the high-intensity CHO compounds with 6-11
oxygen atom were detected in summer, such as $C_{15}H_{24}O_6$, $C_{15}H_{22}O_{10}$, $C_{18}H_{26}O_8$, and $C_{18}H_{26}O_9$,
these highly oxygenated organic molecules with high molecular weight have also been detected in
laboratory α-pinene ozonolysis SOA (Pospisilova et al., 2020). We further classified the CHO
compounds by different carbon atom numbers. As shown in Fig. S6, the $C_{17}$-$C_{22}$ compounds were
the main components of the CHO compounds, accounting for more than 50 % of the total number
of CHO molecular formulas in both summer and winter seasons. However, the total relative
intensities of the CHO compounds in summer were significantly higher than those in winter, of
which the $C_{23}$-$C_{26}$ and $C_{27}$-$C_{32}$ compounds were enrich in summer. These high molecular weight
compounds were probably oligomers formed from various biogenic precursors, such as isoprene,
sesquiterpene, and monoterpene (Daellenbach et al., 2019; Berndt et al., 2018). The high intensities
of these compounds in summer further indicated the stronger biogenic SOA formation in summer
compared with that in winter.

High-intensity CHO compounds with 4-9 oxygen atom were detected in winter (Fig. 5c) of

which the $C_{14}H_{10}O_4$ formula with a DBE value of 10 appeared the highest intensity, which was
probable functional PAHs and have been reported in HULIS from coal combustion smoke particles
(Song et al., 2019). As shown in Fig. S2 and S3, the $C_{14}H_{10}O_4$ formula appeared high intensity in
all the winter samples, providing the evidence of coal combustion emissions in winter. Some other
high-intensity compounds in winter, such as $C_{14}H_8O_4$ and $C_{14}H_8O_5$ both with DBE values of 11,
and $C_{13}H_8O_2$, $C_{13}H_8O_5$, and $C_{13}H_8O_6$ with DBE values of 10, might refer to hydroxyl substitutions
derived from anthracenedione and xanthone, respectively, which have been reported in secondary
wood combustion products (Bruns et al., 2015). $C_{15}H_{10}O_6$, $C_{15}H_8O_6$, and $C_{16}H_{12}O_7$ which had
DBE values of 11, 12, and 11, respectively, might be flavonoids which had flavone backbone, the
key structure of plant pigments, widely existing in plants in nature and could be important sources





of BrC chromophores in aged BBOA (Fleming et al., 2020; Lin et al., 2016; Huang et al., 2021).
Phenolic substances derived from phenol, guaiacol, and syringol are also widely existed in BBOA,
usually from the pyrolysis of lignins in wood, which also play an important role in aqueous-phase
SOA formation (Boreddy et al., 2021). For instance, $C_{13}H_{10}O_3$ and $C_{13}H_{10}O_5$ are guaiacol
derivatives, $C_{15}H_{16}O_8$ are syringol derivatives and $C_{18}H_{14}O_6$ and $C_{18}H_{14}O_7$ are phenol derivatives
(Sun et al., 2021). As shown in Fig. S7, the relative intensities of the CHO compounds mentioned
above produced from BB were found to have similar trends with the mass concentrations of
levoglucosan, which were significantly higher in W1-W3 samples, corresponding to the BB period
from 31 December 2017 to 1 January 2018, providing the evidence of BB influence on HULIS
formation in winter.

It is noted that the top compounds $C_9H_6O_7$ and $C_{10}H_6O_8$ were detected both in the summer

and winter samples (Fig. S2 and S3), which had DBE values of 7 and 8, respectively, containing
abundant condensed aromatic ring structures with high O numbers. Their peaks were also detected
in the HFO (heavy-fuel-oil)-fueled off-road engine samples reported before, suggesting the traffic
emission contributions to HULIS (Cui et al., 2019). This supported the radiocarbon analysis results
in this study and gave further information that the traffic emissions were important fossil sources
in both summer and winter seasons.
3.4.2 Molecular characteristics of CHON compounds

The $O/C_w$ of CHON compounds in summer and winter were 0.57-0.71 and 0.52-0.56,

respectively, while the $H/C_w$ were 1.20-1.32 and 1.00-1.11, respectively (Table S1 and S2).
Compared with the summer CHON compounds, the winter CHON compounds presented
significant higher ion abundance (Fig. 6a and 6d). The most abundant CHON subgroups had DBE
values of 4-7 and 3-10 in summer and winter, respectively (Fig. 6b and 6e). Similar with CHO
compounds, the higher DBE values of high-intensity CHON compounds in HULIS in winter
indicated a high prevalence of double bonds or ring structures. According to the N and O number,
the CHON compounds were classified into $N_1O_x$ ($N_1O_1$-$N_1O_{15}$) and $N_2O_x$ ($N_2O_2$-$N_2O_{14}$) subgroups
in summer and $N_1O_x$ ($N_1O_1$-$N_1O_{12}$) and $N_2O_x$ ($N_2O_2$-$N_2O_{12}$) subgroups in winter, respectively (Fig.
6c and 6f). $NO_{8-12}$ and $NO_{6-9}$ compounds were most enriched subgroups in summer and winter,
respectively. More oxygen-enriched CHON compounds containing O number above 9 were
detected in summer, implying the higher oxidation degree for summer samples. In addition, the
$N_1O_x$ were both the major compounds represented average of $64 \pm 4$ % and $61 \pm 6$ % of the CHON





molecular formulas in summer and winter, respectively, indicating the presence of more single
nitro/amino substituents in CHON compounds in this study.
Among the CHON compounds, $95 \pm 1$ % and $86 \pm 3$ % CHON compounds had O/N values
$\geq 3$ in summer and winter, respectively in this study, indicating these compounds contained large
amounts of oxidized nitrogen functional groups such as nitro compounds ($-NO_2$) and/or
organonitrates ($-ONO_2$) and excess oxygen atoms indicated the existence of other oxygen-
containing functional groups (Laskin et al., 2009). The organonitrates formation from $NO_3$
oxidation of biogenic or anthropogenic VOCs can affect the interactions between anthropogenic
and natural emissions (He et al., 2021; Shen et al., 2021; Wang et al., 2020). Organonitrates were
found to be important species contributing to SOA formation in polluted urban environment, which
were enhanced under high $NO_x$ level (Zheng et al., 2021). The significant higher relative intensities
of CHON compounds in winter indicated that the high $NO_x$ environment in winter promoted the
formation of organonitrates and highlighted the importance of orgnonitrates for SOA control in
polluted environment.
Furthermore, we found that the increase of the relative abundance of CHON compounds in
winter was particularly significant in W1-W3 samples (Fig. S2 and S3), corresponding to the BB
episode. Phenols produced from the pyrolysis of lignins can react with $NO_3$ radicals in the
atmosphere, producing nitrophenols, which have been shown to be important BrC chromophore
in BBOA (Wang et al., 2017b; Lin et al., 2016; Cai et al., 2020). It was reported that the gas-phase
reactions of $NO_3$ radicals with phenolic substances took place at least 4 orders of magnitude faster
than those with aromatic hydrocarbon and even faster in the aqueous phase (Lin et al., 2017).
Among the top CHON compounds with high relative abundance in W1-W3 samples, such as
$C_6H_4N_2O_6$ and $C_7H_6N_2O_6$ with DBE values of 5 and 6, respectively, were refer to nitrophenols
containing one or two nitrogen-containing functional groups, which have been widely reported in
aged BBOA, indicating the increasing of the CHON compounds relative intensity in W1-W3
samples were closely related to BB (Lin et al., 2017; Cai et al., 2020; Mohr et al., 2013; Kourtchev
et al., 2016; Lin et al., 2016). Some other top CHON compounds in winter samples such as
$C_9H_4NO_4$ and $C_{10}H_6NO_4$ with low O/C and H/C ratios most likely indicated the presence of
condensed aromatic structures in the compounds. The $C_9H_4NO_4$ compounds were most likely
emitted from vehicle emissions which have previously been reported (Cui et al., 2019).





It is worth noting that some high-intensity CHON compounds with low O/C and H/C ratios
were detected in summer samples in this study (Fig. 6a), which were closely related to aromatic
compounds from anthropogenic emissions. The top compounds with molecular formulas of
$C_8H_5N_2O_2$ and $C_{19}H_{11}N_2O_4$, which had O/N of 2 and 1, respectively, both were reduced N
compounds referring to N-heterocyclic compounds. Previously study have found that the N-
heterocyclic aromatic compounds can be formed through the aldehyde−ammonia reactions (De
Haan et al., 2018; Zhang et al., 2022). This indicated the important role of reduced N species (e.g.,
ammonium) in the formation of anthropogenic SOA in summer.
3.4.3 Molecular characteristics of S-containing compounds (CHOS and CHONS compounds)
The $O/C_w$ of CHOS compounds in summer and winter were 0.60-0.79 and 0.56-0.67,
respectively, while the $H/C_w$ were 1.50-1.54 and 1.53-1.72, respectively. The $O/C_w$ of CHONS
compounds in summer and winter were 0.82-1.01 and 0.76-0.94, respectively, while the $H/C_w$
were 1.57-1.65 and 1.58-1.66, respectively (Table S1 and S2). As shown in Fig. 7a, 7d, 8a, and 8d,
the high-intensity S-containing compounds in summer and winter were both located at the area
where O/C >0.5 and H/C >1.5, respectively. In addition, the relative intensity of S-containing
compounds increased with the O/C ratios, suggesting the S-containing compounds were highly
oxidized. A small number of high-intensity S-containing compounds with O/C <1.0 and H/C <1.0
were also found in winter in this study, which might be related to OSs and nitrooxy-OSs produced
from the oxidation of aromatic hydrocarbon. The CHOS compounds presenting high relative
abundance were rich in $O_{6-9}S$ and $O_{5-7}S$ groups in summer and winter, respectively, of which the
DBE values were all below 4. The CHONS compounds were rich in $O_{8-10}S$ and $O_{7-9}S$ groups in
summer and winter, respectively, of which the DBE values were all below 6 (Fig. 7b, 7e, 7c, 7f,
8b, 8e, 8c, and 8f). Compared with those of the CHO and CHON compounds, the DBE values of
S-containing compounds were significantly lower.
Among the S-containing compounds, more than 95 % of the CHOS, $CHON_1S$, and $CHON_2S$
formulas had O/S ratios greater than 4, 7, and 10, respectively, implying these compounds may
contain organic sulfate functional groups ($-OSO_3$) or one or two organic nitrate groups ($-ONO_2$)
and these compounds are more likely OSs or nitrooxy-OSs, presenting lower DBE values and
higher O/C and H/C ratios (Table S5 and S6) (O'Brien et al., 2014). The high-intensity CHONS
compounds observed in this study, such as $C_{10}H_{16}NO_{7-9}S$, $C_{10}H_{18}NO_{8-9}S$, $C_{10}H_{18}N_2O_{11}S$, and





$C_9H_{14}NO_8S$ could be nitrooxy-OSs derived from limonene, α-terpinene, and monoterpene (Figure
S2 and S3) (Sun et al., 2021; Bruggemann et al., 2020; Wang et al., 2020; Wang et al., 2018c).
The CHOS compounds with high intensity abundance, such as typical isoprene epoxydiols
(IEPOX) derived OSs with molecular formulas of $C_5H_8O_7S$ and $C_5H_{10}O_7S$ were both detected in
the summer and winter samples, of which the relative intensity of $C_5H_8O_7S$ were over 80 % in S1,
S2, S5, and S6 samples, indicating the significant isoprene SOA formation in summer (Kourtchev
et al., 2016; Kourtchev et al., 2013). The results were consistent with the previous research on the
sources of HULIS based on positive matrix factorization (PMF) model reported by our laboratory
(Bao et al., 2022). The monoterpenes derived OSs such as $C_8H_{14}O_6S$, $C_8H_{14}O_8S$, $C_{10}H_{18}O_8$,
$C_{10}H_{14}O_6$, and $C_{11}H_{16}O_7$ were detected in both summer and winter samples in this study, which
could refer to monoterpene-OSs derived from α-pinene, α-terpinene, and limonene (Wang et al.,
2020). Moreover, OSs with high carbon numbers (C ≥14) such as $C_{14}H_{22}O_7S$, $C_{14}H_{22}O_8S$,
$C_{14}H_{24}O_7S$, $C_{15}H_{26}O_7S$, $C_{15}H_{24}O_7S$, $C_{15}H_{24}O_8S$, and $C_{16}H_{28}O_7S$ were also observed in both
summer and winter samples. Long-chain alkanes emitted from vehicle emissions might be
precursors of these OSs which was consistent with the molecular structures of OSs collected in
urban areas affected by traffic emissions such as Shanghai, Los Angeles, and Beijing (Wang et al.,
2019a; Tao et al., 2014; Wang et al., 2016). The aromatic OSs such as naphthalene derived OSs
with molecular formulas of $C_{10}H_{10}O_6S$, $C_{10}H_{10}O_7S$, and $C_{10}H_{12}O_7S$, 2-methylnaphthalene derived
OSs with molecular formulas of $C_9H_{12}O_6S$, $C_{11}H_{12}O_7S$, and $C_{11}H_{14}O_7S$, and hydroxybenzene
derived OSs with molecular formulas of $C_6H_6O_5S$ were also observed in this study (Qi et al., 2021;
Riva et al., 2015; Blair et al., 2017). Figure S8 further displays the ternary plot of the relative
intensities of OSs from biogenic precursors (e.g., isoprene and monoterpenes), long-chain alkanes
and aromatic hydrocarbon. As shown in Fig. S8, the biogenic OSs and long-chain alkanes OSs
formation were comparable in summer and winter, demonstrating both biogenic and anthropogenic
emission contributions to HULIS. The aromatic OSs presented higher relative intensities in winter,
further indicating the increasing anthropogenic emissions in winter. The presence of long-chain
alkanes derived OSs in both summer and winter seasons provided another evidence that the traffic
emission was one of the important fossil sources of HULIS in this study.
3.5 Comparison with organic compounds in source and atmospheric aerosol samples
The O/C and H/C ratios of water soluble HULIS in this study were compared with those of
water soluble organic compounds reported in source samples from BB, coal combustions, and





vehicle emissions (Tang et al., 2020; Song et al., 2018; Cui et al., 2019; Song et al., 2019), cloud
water samples (Bianco et al., 2018; Zhao et al., 2013), rainwater samples (Altieri et al., 2009), fog
samples (Brege et al., 2018), as well as aerosol samples collected in Beijing (Jang et al., 2020; Wu
et al., 2019a; Wang et al., 2018a), Tianjin (Han et al., 2022), Baoding (Sun et al., 2021), Shanghai
(Wang et al., 2017a), Guangzhou (Jiang et al., 2021), respectively in China, Mainz (Wang et al.,
2018a), Cork city (Kourtchev et al., 2014), and Bologna (Brege et al., 2018), respectively in Europe,
and Bakersfield (O'Brien et al., 2014) and Virginia (Willoughby et al., 2014), respectively in the
United States (Fig. 9). The O/C ratios were obviously higher than those detected in primary BB,
coal combustion, and vehicle emission samples. The H/C ratios of the CHO and CHON
compounds were comparable with the source samples, indicating the organics in HULIS
experienced atmospheric secondary process and the mixed sources of HULIS in this study. The
H/C ratios of the S-containing compounds were much higher than those of source samples which
could be attributed to the significant organosulfates formation in the atmosphere.
The O/C ratios reported in this study were also higher than those reported in aerosol samples
in urban area in China, further indicating the serious secondary pollution at Nanjing, China.
Among the CHO and CHON compounds, we found that the highest H/C ratio values were observed
in the southern city of Guangzhou, followed by those in Nanjing and Shanghai, and the lowest
values were observed in the northern cities such as Beijing, Tianjin, and Baoding, indicating the
higher unsaturation degree of the aerosol samples collected from the northern heating cities, which
were also considered as the heavy industrial region in China. The higher H/C ratios of aerosol
samples collected in Europe and the United States indicated the less anthropogenic emissions such
as industrial emissions from those areas.
**4. Conclusions**
This study focuses on the sources and molecular characteristics differences of water soluble
HULIS in summertime and wintertime from 2017 to 2018 at a suburb site of the YRD, China based
on the radiocarbon analysis and FT-ICR MS measurement with ESI ion source in negative mode.
The carbon isotope analysis results highlight both important fossil and non-fossil source
contributions to HULIS at the study site. A total of 14387 and 15731 peaks were detected in the
summer and winter samples, respectively based on the FT-ICR MS results. The assigned molecular
formulas were classified into CHO, CHON, CHOS, and CHONS subgroups according to their
elemental compositions. The Van Krevelen diagrams showed that more tannins-like and





carbohydrates-like species were detected in summer indicating biogenic SOA formation. Whereas
more compounds containing condensed aromatic structures were detected in winter which were
derived from anthropogenic emissions. The total relative intensity of CHO compounds in summer
were significantly higher than those in winter, containing lots of macromolecular oligomers
derived from biogenic precursors. The high-intensity CHO compounds in winter were mainly
aromatic compounds such as phenolic substances and flavonoids which were related to aged
BBOA and oxidized PAHs most probably from fossil fuel combustion. On the contrary, the total
relative intensity of CHON compounds significantly increased in winter, mainly composed of nitro
compounds or organonitrates. The enhanced formation of nitrophenols in winter indicated the BB
influence. The increasing organonitrates formation in winter highlighted the secondary N-
containing compounds formation via $NO_3$ radical-initiated oxidation processes. It is worth noting
that the top CHON compounds in summer were referring to reduced N compounds produced from
the aldehyde−ammonia reactions. The S-containing compounds were mainly composed of highly
oxidized OSs. The monoterpenes derived OSs and long-chain alkanes derived OSs were widely
observed in both summer and winter samples, while the aromatic OSs formation were found to be
more significant in winter. The presence of long-chain alkanes derived OSs supported the
radiocarbon results, proving that the traffic emission was the important fossil sources at the study
site. Our results highlighted the equal importance of future reduction in both fossil and non-fossil
emissions on atmospheric pollution control.

**Acknowledgments**

This research was financially supported by the National Natural Science Foundation of China

(grant no. 42192512) and the National Natural Science Foundation of China (grant no. 41977305).

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



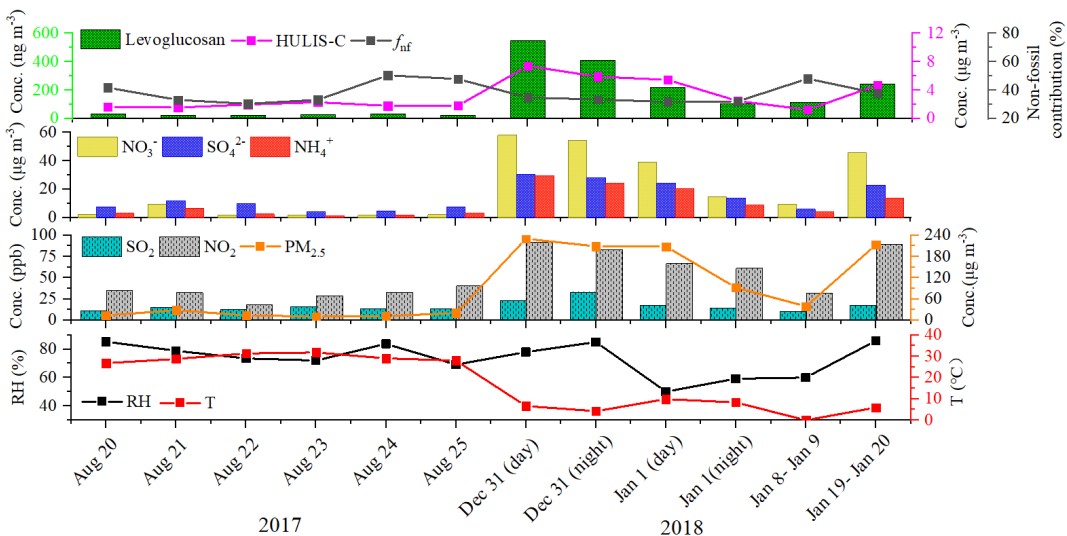

867

Figure 1. Time series of non-fossil contributions to HULIS-C, the mass concentrations of HULIS-

C, Levoglucosan, $NO_3^-$, $SO_4^{2-}$, $NH_4^+$, $SO_2$, $NO_2$, and $PM_{2.5}$, relative humidity, and temperature

during the study periods.



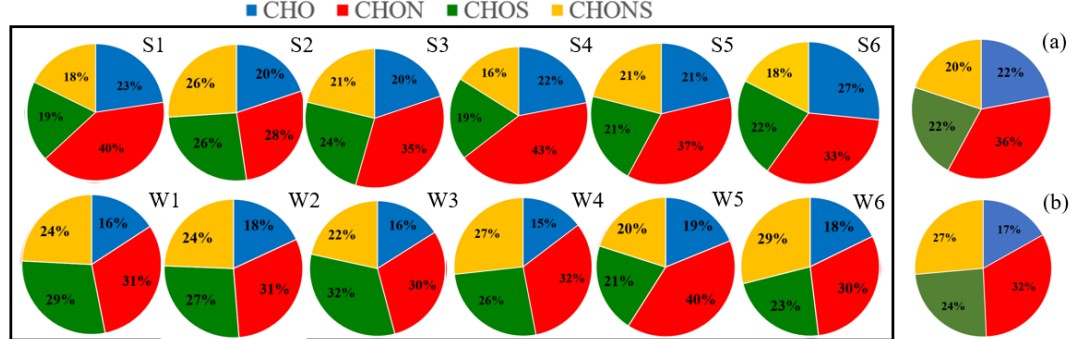

Figure 2. Pie graph of the number percentages of each elemental formula group for the 12 samples plotted in the box and the averaged number percentages of each elemental formula group for the summer samples (a) and winter samples (b).







Figure 3. Van Krevelen diagrams of the 12 samples.






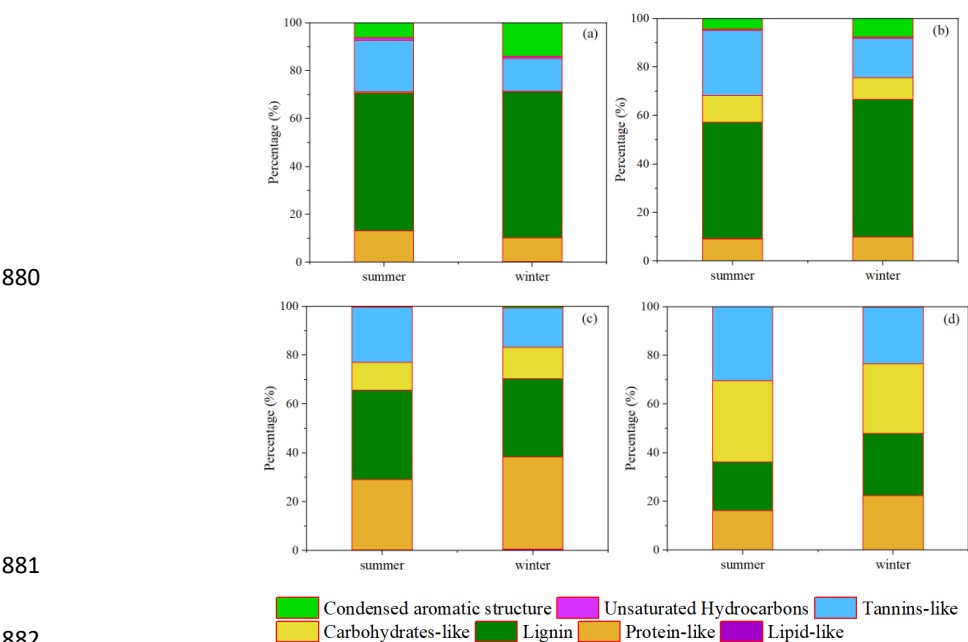

Figure 4. Contributions of seven categories in CHO (a), CHON (b), CHOS (c), and CHONS (d)

compounds.




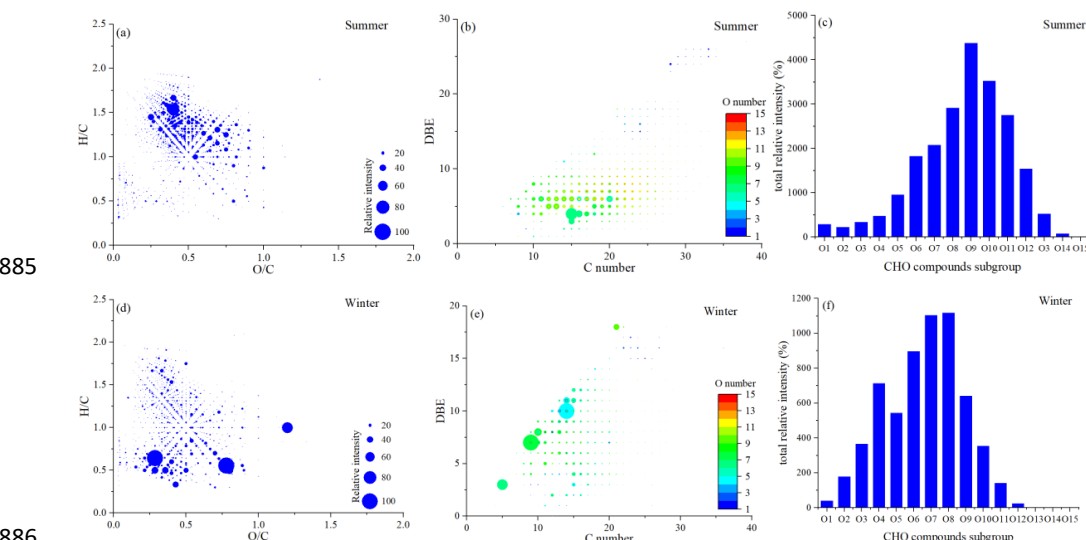


Figure 5. Van Krevelen diagram ((a) and (d)), plot of DBE values vs carbon atom numbers ((b)
and (e)), and the total relative intensity of each subgroup ((c) and (f)) for the CHO compounds in
summer and winter.

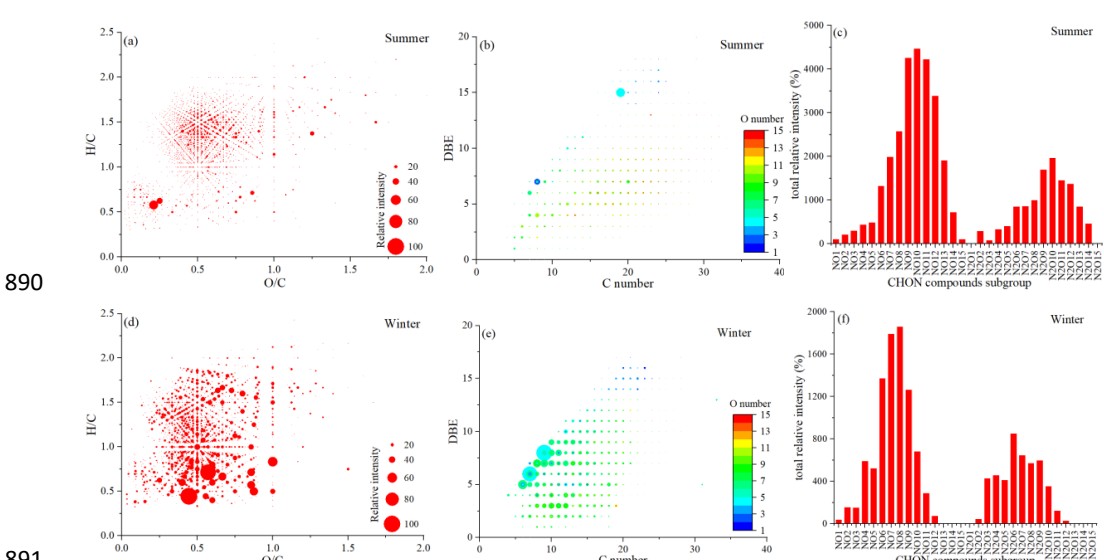



Figure 6. Van Krevelen diagram ((a) and (d)), plot of DBE values vs carbon atom numbers ((b)
and (e)), and the total relative intensity of each subgroup ((c) and (f)) for the CHON compounds
in summer and winter.

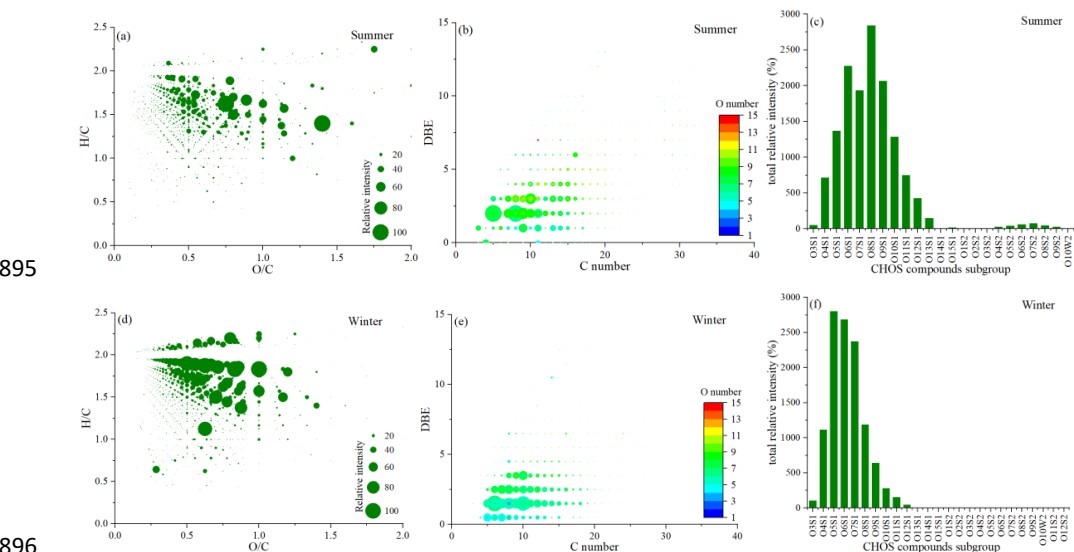

Figure 7. Van Krevelen diagram ((a) and (d)) , plot of DBE values vs carbon atom numbers ((b)
and (e)), and the total relative intensity of each subgroup ((c) and (f)) for the CHOS compounds in
summer and winter.






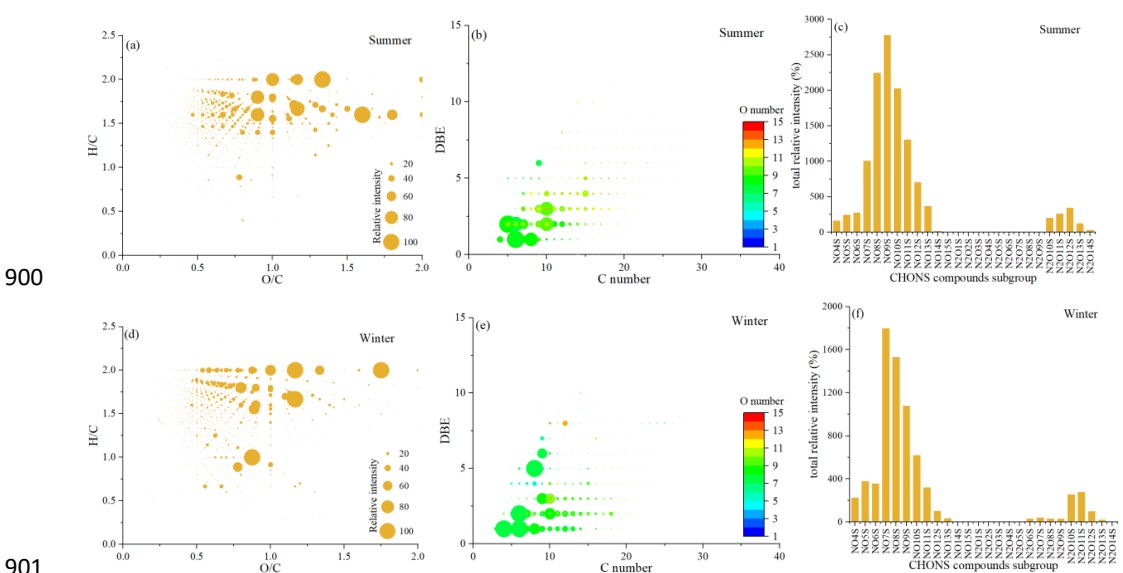


Figure 8. Van Krevelen diagram ((a) and (d)), plot of DBE values vs carbon atom numbers ((b) and (e)), and the total relative intensity of each subgroup ((c) and (f)) for the CHONS compounds in summer and winter.



906

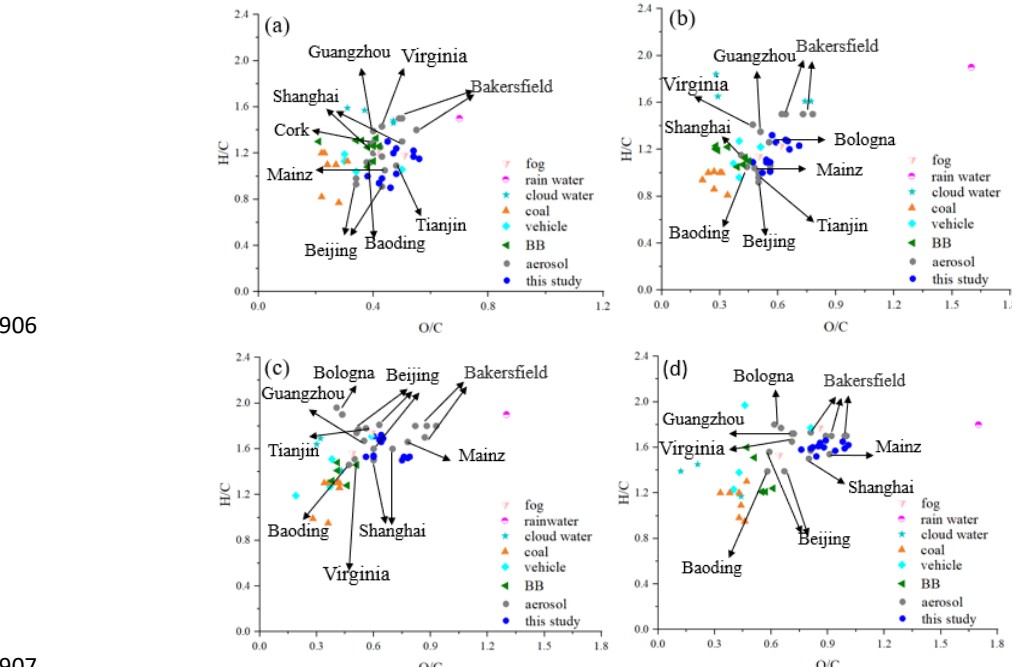

907

Figure 9. Comparison of O/C and H/C ratios of water soluble organic compounds in different atmospheric media in CHO (a), CHON (b), CHOS (c), and CHONS (d) compounds.