# Peer review of "Impact of fossil and non-fossil sources on the molecular compositions of water soluble humic- like substance in PM$_{2.5}$ at a suburb site of Yangtze River Delta, China"

_Atmospheric Chemistry and Physics, 2022_

## Author Comment (AC1)

**Response to reviewer's comments**

We thank the referee for the useful comments and suggestions which have helped us to improve the manuscript. Our point-by-point responses are below. The referee's comments are in black font and our responses are in blue font.

**Referee comments #1**

This study analyzed radiocarbon and molecular composition of HULIS from Nanjing, China. They found both the fossil and non-fossil sources contributed substantially to HULIS. Interestingly, the different patten of the molecular composition in CHO, CHON and CHOS compounds showed that a very different formation mechanism of HULIS in winter and summer. This paper was well organized, written and the method provided in this study was relatively new. I will recommend for a publication after they may address the following comments.

R: We thank the reviewer for the brief summary and positive comments on our work.

Line 147: did the authors analyze the blanks?

R: No field blank correction was performed for the carbon isotope analysis since the carbon content in the field blanks was negligible. However, $^{14}$C analysis of the oxalic acid standard (IAEA-C7), a standard material with $f_m$ value of 0.4953, was conducted in this study and the measured $f_m$ value of IAEA-C7A was 0.4975±0.0018 (Xu et al., 2021). We added the sentence "$^{14}$C analysis of the oxalic acid standard (IAEA-C7) was conducted in this study (Xu et al., 2021)." in the revised manuscript. (See lines 168-169)

Line 165: do S and W mean summer and winter, respectively?

R: Yes. We added the explanation of S1-S6 and W1-W6. The new sentence in the revised manuscript was changed to "The 12 samples were named as S1-S6 (summer) and W1-W6 (winter) in chronological order corresponding to the six samples in summer and winter, respectively in this study." (See lines 187-189)

Line 178: ng or ug for Lev?

R: The unit of the levoglucosan concentrations should be ng m$^{-3}$. We corrected the mistake. (See line 214)

Line 182: I think fossil sources were even more important from the $^{14}$C analysis. So this should be stated.

R: We thank the reviewer for pointing out the issue. We adjusted the descriptions in the revised manuscript to "The radiocarbon analysis results showed that the $f_{nf}$ of HULIS-C ranged from 30 % to 50 % with an average contribution of 39 ± 8 % in summer and ranged from 32 % to 48 % with an average contribution of 36 ± 6 % in winter, indicating the significant contributions from fossil sources to HULIS at the study site." (See lines 205-209)

We also complemented the descriptions in the abstract and conclusions to support the results for the important fossil source contributions found in this study. At the end of the abstract, we stressed that "Generally, different policies need to be considered for each season due to the different season sources, i.e., biogenic emissions in summer and biomass burning in winter for non-fossil sources, traffic emission and anthropogenic SOA formation in both seasons and additional coal combustion in winter. Measures to control emissions from motor vehicles and industrial processes need to be considered in summer. Additional control measures on coal power plants and biomass burning should be concerned in winter." (See lines 46-52)

In the conclusion section in the revised manuscript, we pointed out that "The presence of long-chain alkanes derived OSs supported the radiocarbon results, indicating that the traffic emission was the important fossil sources at the study site. The presence of aromatic secondary N-containing and S-containing compounds provided evidence for the substantial contributions from anthropogenic SOA formation to fossil sources at the study site." (See lines 563-567)

Line 182-183: this sentence was not clear.

R: We removed the sentence considering this sentence was not described logically here and we added more explanation here which was shown in the next comment.

Line 185-186: the logic was not clear, and more explanation was needed.

R: We adjusted the logic and added new descriptions to explain the sources in summer and winter, respectively from the back trajectories analysis. "The 48 h back trajectories (Fig. S1) showed that the study site was affected by the polluted air masses mainly from the northern cities in winter, suggesting the coal combustion contributions to HULIS in winter. In addition, significant increasing of the levoglucosan and HULIS-C mass concentrations were found from 31 December 2017 to 1 January 2018, corresponding to the W1-W3 samples and the maximum of the levoglucosan and HULIS-C mass concentrations were 552.79 ng m$^{-3}$ and 7.40 µg m$^{-3}$, respectively, indicating the importance of the biomass burning contribution. In summer, the study site was affected by both regional transport from the nearby cities in the north and west of Nanjing and the

Donghai Sea. The anthropogenic emissions from the neighboring cities might cause the anthropogenic SOA formation, i.e., secondary N-containing and S-containing compounds with aromatic structures during the atmospheric transport processes, which was discussed in detail in section 3.4 in this study." (See lines 209-220)

Line 228: what did the higher values indicate?

R: The higher $DBE_w$ and $DBE/C_w$ values of CHO and CHON compounds indicated the higher unsaturation degree of these two groups. We added the explanation in the revised manuscript. (See lines 272-273)

Line 293-297: it seemed that the summary may not fully represent the findings.

R: We thank the reviewer for pointing out the issue. We added more contents in the summary. The new summary was shown as "In summary, both the summer and winter samples were mainly composed of compounds from biogenic origins (lignins-like, tannins-like, proteins-like, and carbohydrates-like species). More tannins-like and carbohydrates-like species were detected in summer including large amounts of highly oxidized OSs or nitrooxy-OSs, indicating biogenic SOA formation. More condensed aromatic structures in CHO and CHON compounds were detected in winter, owing to increasing anthropogenic emissions." (See lines 338-343)

Figure 1 should add S1-S6 or W1-W6 in the figure. Or add a table

R: We added the corresponding S1-S6 and W1-W6 in Figure 1 in the revised manuscript.

**References:**

Xu, B., Cheng, Z., Gustafsson, Ö., Kawamura, K., Jin, B., Zhu, S., Tang, T., Zhang, B., Li, J., and Zhang, G.: Compound-specific radiocarbon analysis of low molecular weight dicarboxylic acids in ambient aerosols using preparative gas chromatography: method development, Environ. Sci. Technol. Lett., 8, 135-141, 10.1021/acs.estlett.0c00887, 2021.

---

## Author Comment (AC2)

**Response to reviewer's comments**

We thank the referee for the useful comments and suggestions which have helped us to improve the manuscript. Our point-by-point responses are below. The referee's comments are in black font and our responses are in blue font.

General comments

The manuscript investigates fossil and non-fossil sources' contribution to the molecular composition of water-soluble humic-like substances (HULIS) in $PM_{2.5}$ during the summer and winter in Nanjing. In addition, the study employed radiocarbon and chemical analyses to characterize water-soluble HULIS molecular composition and to examine the different sources' influence on the molecular composition.

The authors discuss the molecular compositions and their seasonal difference. The seasonal difference was found to be originated from seasonal sources, i.e., biomass burning in the winter and biogenic emission in the summer. The study concludes that fossil and non-fossil sources are equally important in the air pollution reduction policy. I found the conclusion is overreaching. However, the results presented don't provide information on the equal contributions of fossil and non-fossil sources to the HULIS composition. This issue, along with others provided below, needs to be addressed.

R: We thank the reviewer for pointing out the issue and have answered this question together with others in the specific comments below.

Overall, the study is well within the scope of the Atmospheric Chemistry and Physics journal and is a good addition to the current knowledge. The manuscript flows logically. I found a few technical issues, as listed below. Finally, I recommend publishing the manuscript after addressing the comments.

R: We thank the reviewer for the brief summary and positive comments on our work.

Specific comments

1、Line 512 suggests the equal importance of fossil and non-fossil fuels in atmospheric aerosol in Nanjing. The fraction of non-fossil (f-nf) of HULIS-C is, on average, $39 \pm 8$ % in the summer and $36 \pm 6$ % in the winter. That means they are not equally contributing (+/- 50% : 50%) to HULIS in $PM_{2.5}$ and air pollution in general. Moreover, there is a slight difference due to seasonal sources, i.e., biogenic emission and biomass burning. That infers that the policy needs to consider the types of pollution for each season instead of generalizing the contribution of the sources. This issue needs to be addressed and clarified.

R: We changed the sentence in section 3.1 to "the radiocarbon analysis results showed that the $f_{nf}$ of HULIS-C ranged from 30 % to 50 % with an average contribution of $39 \pm 8$ % in summer and ranged from 32 % to 48 % with an average contribution of $36 \pm 6$ % in winter, indicating the significant contributions from fossil sources to HULIS at the study site." We further supported the results by adding more explanations here. "The 48 h back trajectories (Fig. S1) showed that the study site was affected by the polluted air masses mainly from the northern cities in winter, suggesting the coal combustion contributions to HULIS in winter. In addition, significant increasing of the levoglucosan and HULIS-C mass concentrations were found from 31 December 2017 to 1 January 2018, corresponding to the W1-W3 samples and the maximum of the levoglucosan and HULIS-C mass concentrations were 552.79 ng m$^{-3}$ and 7.40 μg m$^{-3}$, respectively, indicating the biomass burning impact during the periods. In summer, the study site was affected by both regional transport from the nearby cities in the north and west of Nanjing and the Donghai Sea. The anthropogenic emissions from the neighboring cities might cause the anthropogenic SOA formation, i.e., secondary N-containing and S-containing compounds with aromatic structures during the atmospheric transport processes, which was discussed in detail in section 3.4 in this study." (See lines 205-220)

We thank the reviewer for pointing out the issue on the different sources of HULIS, i.e., biogenic emission in summer and biomass burning in winter. At the end of the abstract, we pointed out that "Generally, different policies need to be considered for each season due to the different season sources, i.e., biogenic emission in summer and biomass burning in winter for non-fossil source, traffic emission and anthropogenic SOA formation in both seasons and additional coal combustion in winter. Measures to control emissions from motor vehicles and industrial processes need to be considered in summer. Additional control measures on coal power plants and biomass burning should be concerned in winter." (See lines 46-52)

2、The introduction needs more context on the study location. What has been found so far in Nanjing, what needs to be added to the current knowledge, and how does this study fit/enrich the knowledge?

R: We added new contents in the introduction section in the revised manuscript to describe the current researches at the study location. Organic matter can account for 20-40 % of $PM_{2.5}$ in the Yangtze River Delta area due to the impact of complicated sources, especially anthropogenic emissions (Wang et al., 2017; Wang et al., 2016). Studies have reported that Brown carbon (BrC) is an important contributor to aerosol light absorption in Nanjing and exhibited obvious seasonal variations, with peaks in wintertime, owing to emissions from biomass burning, fossil fuel combustion, and secondary formation (Chen et al., 2018; Cui et al., 2021; Xie et al., 2020; Wang et al., 2018). Recently, works on the field observation of nitrated aromatic compounds (NACs) were conducted to explore the light absorption contributions of NACs to BrC and help to better understand the links between the optical properties and molecular compositions of BrC (Gu et al., 2022; Cao et al., 2023). However, as far as we know, understanding of the sources of atmospheric HULIS at molecular levels was still limited. In this work, we aim to obtain the molecular characteristic differences of water soluble HULIS in summertime and wintertime and to get a better understanding of the influence of different sources on the molecular compositions of HULIS. (See lines 107-117)

3、Some clarifications for the following.

- What parameters are provided by China Environmental Monitoring Centre (Line 121)? Are the parameters the ones mentioned in Line 163?

R: The air pollutants data including $PM_{2.5}$, $SO_2$ and $NO_2$ were provided by China National Environmental Monitoring Centre. We listed the parameters in the revised manuscript. (See line 136)

- Add a brief description of the isolation and measurement of the chemical analysis before referring to previous studies/references (Line 132).

R: We added a brief description of the isolation and measurement of HULIS, as well as the water soluble ions and levoglucosan. (See lines 140-144 and lines 149-155)

- Line 168 mentions NAAQS for the first grade. I don't know "the first grade", but the phrase may not be necessary as the NAAQS level is sufficient. If that is important to mention, add some context.

R: We removed "the first grade" in the revised manuscript. (See lines 193-194)

4、Lines 435-442 discuss the abundant CHOS compounds are isoprene SOA tracers that formation is higher in the summer. Isoprene SOA enhancement in the summer is well understood. Other than enhanced isoprene SOA, what are the consistent results between this study and the previous one (Bao et al., 2022)?

R: According to the previous study reported by Bao et al. (2022), good correlations were found between HULIS with levoglucosan and $K^+$ in winter, suggesting the biomass burning influence on HULIS in winter, which was further supported by the positive matrix factorization (PMF) analysis. Consistent results were found in this study. In addition, the PMF results showed that anthropogenic SOA formation and fossil fuel combustions including industrial and traffic emissions significantly contributed to HULIS in summer (51%) and winter (51%), respectively. Significant fossil sources were found based on both the PMF model before and $^{14}$C analysis in this study. It should be noted that the fossil source contributions derived from the PMF model were a little lower than the $^{14}$C results. The PMF model results were influenced by the input parameters, which possessed definite subjectivity and had uncertainties due to the measurement errors. The $^{14}$C analysis is a powerful tool for distinguishing and determining fossil and non-fossil sources of carbonaceous aerosols due to the fact that all $^{14}$C atoms have completely decayed in fossil fuels, whereas non-fossil sources contain constant $^{14}$C levels (Mo et al., 2018; Liu et al., 2018).

In this study, two high-intensity CHO compounds containing condensed aromatic ring structures ($C_9H_6O_7$ and $C_{10}H_5O_8$) identified in summer and winter samples were similar to those from off-road engine samples, indicating that traffic emission was one of the important fossil sources of HULIS at the study site. The presence of long-chain alkanes derived OSs supported the radiocarbon results, providing another evidence that the traffic emission was the important fossil sources at the study site. The presence of aromatic secondary N-containing and S-containing compounds provided evidence for the anthropogenic SOA formation contribution to fossil sources at the study site. These results further verified the work reported before by Bao et al. (2022) at molecular level and help a better understanding of the interaction between the sources and the molecular compositions of atmospheric HULIS. We added these descriptions in the revised manuscript in lines 407-408 and lines 563-569.

Technical comments

1、Line 63: Do you mean "increase aerosol hygroscopicity"?

R: Yes. We thank the reviewer for finding out the issue. We changed the word "produced" to "increased" in the revised manuscript. (See line 68)

2、Line 398: Both species have a DBE of 6. Unless there's a typo on the Molecular Formula.

R: We thank the reviewer for pointing out the mistake. We changed the sentence to "such as $C_6H_4N_2O_6$ and $C_7H_6N_2O_6$ both with a DBE value of 6". (See line 445)

3、Line 406: Reverse the O/N ratios: "... 1 and 2, respectively".

R: Corrected. (See line 456)

4、Line 407: Typo "Previously, studies have found...".

R: Corrected. (See line 457)

5、Figure 4: I have a minor suggestion. It is difficult to identify the unsaturated HC and lipid-like groups because the bar line is red, which is close to the colors used for those groups. I'd suggest changing the bar line to transparent/no-line or white, similar to Fig. 2 pie charts.

R: We thank the reviewer for the useful suggestion. We changed the bar line to no-line in Figure 4 in the revised manuscript.

**References:**

Bao, M., Zhang, Y. L., Cao, F., Lin, Y. C., Hong, Y., Fan, M., Zhang, Y., Yang, X., and Xie, F.: Light absorption and source apportionment of water soluble humic-like substances (HULIS) in $PM_{2.5}$ at Nanjing, China, Environ. Res., 206, 112554, 10.1016/j.envres.2021.112554, 2022.

Cao, M., Yu, W., Chen, M., and Chen, M.: Characterization of nitrated aromatic compounds in fine particles from Nanjing, China: Optical properties, source allocation, and secondary processes, Environ. Pollut., 316, 120650, 10.1016/j.envpol.2022.120650, 2023.

Chen, Y., Ge, X., Chen, H., Xie, X., Chen, Y., Wang, J., Ye, Z., Bao, M., Zhang, Y., and Chen, M.: Seasonal light absorption properties of water-soluble brown carbon in atmospheric fine particles in Nanjing, China, Atmos. Environ., 187, 230-240, 10.1016/j.atmosenv.2018.06.002, 2018.

Cui, F., Pei, S., Chen, M., Ma, Y., and Pan, Q.: Absorption enhancement of black carbon and the contribution of brown carbon to light absorption in the summer of Nanjing, China, Atmos. Pollut. Res., 12, 480-487, 10.1016/j.apr.2020.12.008, 2021.

Gu, C., Cui, S., Ge, X., Wang, Z., Chen, M., Qian, Z., Liu, Z., Wang, X., and Zhang, Y.: Chemical composition, sources and optical properties of nitrated aromatic compounds in fine particulate

matter during winter foggy days in Nanjing, China, Environ. Res., 212, 113255, 10.1016/j.envres.2022.113255, 2022.

Liu, J., Mo, Y., Ding, P., Li, J., Shen, C., and Zhang, G.: Dual carbon isotopes ($^{14}$C and $^{13}$C) and optical properties of WSOC and HULIS-C during winter in Guangzhou, China, Sci. Total. Environ., 633, 1571-1578, 10.1016/j.scitotenv.2018.03.293, 2018.

Mo, Y., Li, J., Jiang, B., Su, T., Geng, X., Liu, J., Jiang, H., Shen, C., Ding, P., Zhong, G., Cheng, Z., Liao, Y., Tian, C., Chen, Y., and Zhang, G.: Sources, compositions, and optical properties of humic-like substances in Beijing during the 2014 APEC summit: Results from dual carbon isotope and Fourier-transform ion cyclotron resonance mass spectrometry analyses, Environ. Pollut., 239, 322-331, 10.1016/j.envpol.2018.04.041, 2018.

Wang, J., Ge, X., Chen, Y., Shen, Y., Zhang, Q., Sun, Y., Xu, J., Ge, S., Yu, H., and Chen, M.: Highly time-resolved urban aerosol characteristics during springtime in Yangtze River Delta, China: insights from soot particle aerosol mass spectrometry, Atmos. Chem. Phys., 16, 9109–9127, https://doi.org/10.5194/acp-16-9109-2016, 2016.

Wang, J., Nie, W., Cheng, Y., Shen, Y., Chi, X., Wang, J., Huang, X., Xie, Y., Sun, P., Xu, Z., Qi, X., Su, H., and Ding, A.: Light absorption of brown carbon in eastern China based on 3-year multi-wavelength aerosol optical property observations and an improved absorption Ångström exponent segregation method, Atmos. Chem. Phys., 18, 9061-9074, 10.5194/acp-18-9061-2018, 2018.

Wang, J., Zhao, B., Wang, S., Yang, F., Xing, J., Morawska, L., Ding, A., Kulmala, M., Kerminen, V.-M., Kujansuu, J., Wang, Z., Ding, D., Zhang, X., Wang, H., Tian, M., Petäjä, T., Jiang, J., and Hao, J.: Particulate matter pollution over China and the effects of control policies, Sci. Total. Environ., 584–585, 426–447, 10.1016/j.scitotenv.2017.01.027, 2017.

Xie, X., Chen, Y., Nie, D., Liu, Y., Liu, Y., Lei, R., Zhao, X., Li, H., and Ge, X.: Light-absorbing and fluorescent properties of atmospheric brown carbon: A case study in Nanjing, China, Chemosphere, 251, 126350, 10.1016/j.chemosphere.2020.126350, 2020.

---

## Author Comment (AC3)

**Response to reviewer's comments**

We thank the referee for the useful comments and suggestions which have helped us to improve the manuscript. Our point-by-point responses are below. The referee's comments are in black font and our responses are in blue font.

Based on the radiocarbon analysis combining the Fourier Transform Ion Cyclotron Resonance Mass Spectrometry technique, this study reports the potential fossil and non-fossil sources and molecular compositions of water soluble humic-like at a suburb site of Yangtze River Delta, China. The research was carried out to better understand the interaction between the sources and the molecular compositions of atmospheric HULIS. There are several issues that need to be point out before publication (see additional comments):

Additional comments:

1、Line 60-62: Some recent literatures on the composition of HULIS should be cited here, such as: Optical properties, molecular characterizations, and oxidative potentials of different polarity levels of water-soluble organic matters in winter $PM_{2.5}$ in six China's megacities. Sci. Total. Environ., 853 (2022), 158600; Seasonal and diurnal variation of $PM_{2.5}$ HULIS over Xi'an in Northwest China: Optical properties, chemical functional group, and relationship with reactive oxygen species (ROS). Atmos. Environ., 268 (2022), 118782.

R: We thank the reviewer for providing the recent literatures. We added the provided references in the revised manuscript. (See line 66 and lines 959-966)

2、Line 76: The "lingin-derived products" refers to lignin-derived products?

R: Corrected. (See line 81)

3、Line 177-178: How about the difference between the concentration of HULIS-C in this study and previous literature? For example, HULIS-C in Xi'an can compare with this study. ($PM_{2.5}$ Humic-like substances over Xi'an, China: Optical properties, chemical functional group, and source identification. Atmos. Res., 234 (2020), 104784).

R: We added the description about the comparison between our results with those measured in other cities in China. The details were shown as "The averaged mass concentrations of HULIS in summer and winter during the selected periods were $1.83 \pm 0.27$ μg m$^{-3}$ and $4.52 \pm 2.29$ μg m$^{-3}$,

respectively. The averaged HULIS concentration in summer was comparable with those measured in other cities in China, i.e., 1.70 μg m$^{-3}$ in Guangzhou,1.61 μg m$^{-3}$ in Shanghai and 1.50 μg m$^{-3}$ in Xi'an. Compared with those measured in winter samples in other cities, our result was comparable with those in Xi'an (4.50 μg m$^{-3}$), a little lower than those in the megacity of Shanghai (5.31 μg m$^{-3}$) and higher than those in the southern coastal city of Guangzhou (3.6 μg m$^{-3}$)." (Fan et al., 2016; Zhang et al., 2020; Zhao et al., 2016). (See lines 195-202)

4、Line 185-186: "··· mainly from the northern heating cities in winter, suggesting the coal combustion contributions to HULIS in winter". Additional references are needed here.

R: We added two references here which focus on HULIS in the northern cities in China and found that coal combustions was one of the important sources of HULIS to support our results. (See line 211, lines 771-773 and lines 841-843)

5、Line 191-195: "Our results are similar to those found for the ultrahigh resolution mass spectra of water-soluble organic···". Please explain what does the similarity of these mass spectra indicate?

R: Thousands of peaks were detected for each sample in this study, suggesting the chemical complexity of HULIS. Here we presented the similarity of our samples with those from source samples, ambient aerosols and cloud water samples to illustrate that the data were within a reasonable range. We added the illustration at the end of the sentence. (See line 228)

6、Line 205-208: The source profiles of molecules of HULIS from biomass burning can be used here to indicate the biomass burning contribution. (Light absorption properties and molecular profiles of HULIS in PM$_{2.5}$ emitted from biomass burning in traditional "Heated Kang" in Northwest China. Sci. Total. Environ., 776 (2021), 146014). Are biomass combustion emissions also responsible for the increased contribution of S-containing compounds in winter in this study?

R: We thank the reviewer for providing the references. We added the new references in the revised manuscript. Song et al. (2018) also found that primary HULIS emitted from biomass burning contain a high abundance of CHON compounds and S-containing compounds were the dominant component (78.1%) for HULIS in coal-smoke particles. We changed the sentences to "The CHON groups were the major components of molecular formulas, furthermore, the relative intensity of CHON groups increased significantly in winter (Fig. S2 and Fig. S3). Studies have suggested that HULIS emitted from biomass burning can produce a high abundance of CHON compounds and S-containing compounds were the dominant component for primary HULIS emitted from coal

combustion (Zhang et al., 2021; Song et al., 2018). The higher intensity of CHON compounds in winter in this study further indicated the biomass burning contribution. The contributions of S-containing compounds (CHOS and CHONS groups) increased in winter which might be related to the polluted air masses transported from the northern cities with increasing coal combustions emissions in winter (Song et al., 2018)." (See lines 239-248)

In section 3.4.3, we discussed the molecular characteristics of S-containing compounds in this study. We found the S-containing compounds were mainly composed of organosulfates (OSs) derived from biogenic precursors, long-chain alkane and aromatic hydrocarbon. In addition, the aromatic OSs presented higher relative intensities in winter. However, as far as we know, we could not make sure what's the sources of these OSs. What we can explain was the aromatic OSs in winter indicated the increasing anthropogenic emissions.

7、Line 202-206: As I have seen, the average proportions of CHO and CHON groups were higher in summer (22% and 36%, respectively) than in winter (17% and 32%, respectively). Please explain the possible reasons for this phenomenon.

R: In section 3.4.1 and 3.4.2, we discussed the molecular characteristics of CHO and CHON compounds, respectively. Large amounts of biogenic SOA species were found in summer CHO compounds, indicating the significant biogenic emissions in summer. For the CHON compounds, we found they were mainly composed of organonitrates or nitro compounds with O/N values ≥3. The increasing of biogenic emissions might cause the increasing formation of organonitrates which could be from $NO_3$ oxidation of biogenic VOCs. Above all, we thought the possible reason for the higher proportion of CHO and CHON groups in summer was the increasing biogenic emissions in summer.

We added the sentence in the revised manuscript to explain the higher proportions of CHO and CHON groups in summer at the end of the paragraph which was described as "Notably, the relatively higher proportions of CHO and CHON groups in summer were most probably related to the increasing biogenic emissions in summer, resulting in the formation of some high molecular weight oligomers or highly oxidized organonitrates, which was discussed in detail in section 3.4.1 and 3.4.2 in this study." (See lines 248-251)

8、Line 227-228: What is the difference between $DBE_w$ and $DBE/C_w$? In addition, what does the higher DBE and DBE/C values of CHO and CHON compounds indicate in this study?

R: DBE was the double-bond equivalent of compounds and DBE/C was the double-bond equivalent of unit carbon. We added the description of DBE/C and the indication of higher DBE and DBE/C values of CHO and CHON compounds in the revised manuscript. It was shown as "The related parameter DBE/C was the double-bond equivalent of unit carbon which can reflect the condensed ring structures in the compounds (Jiang et al., 2021). The higher $DBE_w$ and $DBE/C_w$ values of CHO and CHON compounds were found in this study, indicating the higher unsaturation degree of these two groups." (See lines 270-273)

9、In section 3.4, when discussed the formation of HULIS molecular, a recent study on BrC molecular formation can be used to support author conclusion.

The Roles of N, S, and O in Molecular Absorption Features of Brown Carbon in $PM_{2.5}$ in a Typical Semi-Arid Megacity in Northwestern China. Journal of Geophysical Research-Atmospheres 126.

R: We added the references at the end of section 3.4.2 to support our results on the reduced N compounds formation in summer. The details were described as "Our results were consistent with previous study conducted in Xi'an, China which also found formation of reduced N compounds in light-absorbing aerosols through ammonia involved reactions in summer (Zeng et al., 2021)." (See lines 460-462)

10、Line 431-434: "The high-intensity CHONS compounds observed in this study, such as $C_{10}H_{16}NO_{7-9}S$, $C_{10}H_{18}NO_{8-9}S$, $C_{10}H_{18}N_2O_{11}S$…". What are the differences in the abundance of these compounds in winter and summer in this study?

R: We compared the relative intensity of these compounds in summer and winter and the results were shown in Table 1 below. Some of them were higher in summer and some were higher in winter, indicating both biogenic emission contributions to HULIS in summer and winter, respectively, which was also found in CHOS compounds in this study. Despite this, some information still could be found from table 1, for instance, the compounds detected in summer contained more oxygen atoms, indicating the higher oxidation degree of these nitrooxy-OSs in summer. This information can also be seen from Fig. S2 and S3. We changed the sentence to "The high-intensity CHONS compounds observed in this study, such as $C_{10}H_{16}NO_{7-9}S$, $C_{10}H_{18}NO_{8-9}S$, $C_{10}H_{18}N_2O_{11}S$, and $C_9H_{14}NO_{8-9}S$ could be nitrooxy-OSs derived from monoterpenes such as limonene and α-terpinene of which we found the formulas in summer contained more oxygen

atoms, indicating the higher oxidation degree of these nitrooxy-OSs in summer (Figure S2 and S3)." (See lines 485-487)

Table 1. Averaged relative intensity (%) of the high-intensity CHONS compounds in summer and winter, respectively.

|  | Summer | winter |
|---|---|---|
| $C_{10}H_{16}NO_7S$ | 13.66 | 14.74 |
| $C_{10}H_{16}NO_8S$ |  | 21.68 |
| $C_{10}H_{16}NO_9S$ | 56.00 | 26.83 |
| $C_{10}H_{18}NO_8S$ |  | 44.39 |
| $C_{10}H_{18}NO_9S$ | 61.85 | 47.81 |
| $C_{10}H_{18}N_2O_{11}S$ |  | 31.79 |
| $C_9H_{14}NO_8S$ |  | 39.97 |
| $C_9H_{14}NO_9S$ | 37.72 | 20.70 |

**References:**

Fan, X., Song, J., and Peng, P. a.: Temporal variations of the abundance and optical properties of water soluble Humic-Like Substances (HULIS) in $PM_{2.5}$ at Guangzhou, China, Atmos. Res., 172-173, 8-15, 10.1016/j.atmosres.2015.12.024, 2016.

Jiang, H., Li, J., Sun, R., Tian, C., Tang, J., Jiang, B., Liao, Y., Chen, C. E., and Zhang, G.: Molecular dynamics and light absorption properties of atmospheric dissolved organic matter, Environ. Sci. Technol., 55, 10268-10279, 10.1021/acs.est.1c01770, 2021.

Song, J., Li, M., Jiang, B., Wei, S., Fan, X., and Peng, P.: Molecular characterization of water-soluble humic like substances in smoke particles emitted from combustion of biomass materials and coal using Ultrahigh-resolution electrospray ionization fourier transform ion cyclotron resonance mass spectrometry, Environ. Sci. Technol., 52, 2575-2585, 10.1021/acs.est.7b06126, 2018.

Zeng, Y., Ning, Y., Shen, Z., Zhang, L., Zhang, T., Lei, Y., Zhang, Q., Li, G., Xu, H., Ho, S. S. H., and Cao, J.: The roles of N, S, and O in molecular absorption features of brown carbon in $PM_{2.5}$ in a typical semi-arid megacity in Northwestern China, J. Geophys. Res: Atmospheres, 126, 10.1029/2021jd034791, 2021.

Zhang, T., Shen, Z., Zhang, L., Tang, Z., Zhang, Q., Chen, Q., Lei, Y., Zeng, Y., Xu, H., and Cao, J.: $PM_{2.5}$ Humic-like substances over Xi'an, China: Optical properties, chemical functional group, and source identification, Atmos. Res., 234, 10.1016/j.atmosres.2019.104784, 2020.

Zhang, T., Shen, Z., Zeng, Y., Cheng, C., Wang, D., Zhang, Q., Lei, Y., Zhang, Y., Sun, J., Xu, H., Ho, S. S. H., and Cao, J.: Light absorption properties and molecular profiles of HULIS in $PM_{2.5}$ emitted from biomass burning in traditional "Heated Kang" in Northwest China, Sci. Total. Environ., 776, 146014, 10.1016/j.scitotenv.2021.146014, 2021.

Zhao, M., Qiao, T., Li, Y., Tang, X., Xiu, G., and Yu, J. Z.: Temporal variations and source apportionment of Hulis-C in $PM_{2.5}$ in urban Shanghai, Sci. Total. Environ., 571, 18-26, 10.1016/j.scitotenv.2016.07.127, 2016.

---

## Author Response (AR2)

**Response to editor's comments**

We thank the editor for the detailed comments which have helped us to improve the manuscript. Our point-by-point responses are below. The editor's comments are in black font and our responses are in blue font.

There are several minor issues that need to be point out before publication (see additional comments).

Line 88-89: Here, related references needed to be added.

R: We added references in the revised manuscript. (See line 84)

Line 157-159: In this manuscript, "The mass concentrations of ... NO3-, NH4+., and SO42-.were measured using an ion chromatography separated on an AS11 column." AS11 column is an anion column in ion chromatography, but NH4+ is also separated by this column in this manuscript, please check this sentence.

R: We thank the editor for pointing out the issue. We changed the sentence in the revised manuscript to "The mass concentrations of the water soluble ions including $NO_3^-$, $NH_4^+$ and $SO_4^{2-}$ were measured using an ion chromatography (Dionex ICS-5000+, ThermoFisher Scientific, USA) separated on an AS11 column (4*250 mm, Dionex) for anions and a CS12A column (4*250 mm, Dionex) for cations, respectively. Potassium hydrate (KOH) and methane sulfuric acid (MSA) were used as the gradient eluent for anion and cation determination, respectively." (See lines 150-154)

Line 215: Avoid lumping references as in (Fan et al., 2016; Zhang et al., 2020b; Zhao et al., 2016) and all other. Instead summarize the main contribution of each referenced paper in a separate sentence.

R: We changed the sentences in the revised manuscript to "Compared with those measured in other cities in China in summer, the averaged HULIS concentration in Nanjing in summer was comparable with those measured in Guangzhou of 1.70 µg m$^{-3}$ (Fan et al., 2016), Shanghai of 1.61 µg m$^{-3}$ (Zhao et al., 2016) and Xi'an of 1.50 µg m$^{-3}$ (Zhang et al., 2020b). Compared with those measured in winter samples in other cities, our result was comparable with those in Xi'an of 4.50 µg m$^{-3}$ (Zhang et al., 2020b), a little lower than those in the megacity of Shanghai of 5.31 µg m$^{-3}$

(Zhao et al., 2016) and higher than those in the southern coastal city of Guangzhou of 3.60 μg m$^{-3}$ (Fan et al., 2016)." (See lines 200-206)

Line 304-305:"proposed by previous studies", related references needed to be added.

R: We added references in the revised manuscript. (See line 286)

Line 366-367: Please supplement references to prove that "more sensitive for the identification of polar compounds".

R: We added references in the revised manuscript. (See line 349)

Line 539-548: This sentence is too long.

R: We divided the sentence into two new sentences in the revised manuscript. (See line 520-530)